# FITTING NETWORKS WITH A CANCELLATION TRICK

**Jiashun Jin**
Department of Statistics
Carnegie Mellon University
Pittsburgh, PA 15213, USA
jiashun@andrew.cmu.edu

**Jingming Wang**
Department of Statistics
University of Virginia
Virginia, VA 22903, USA
pdw9qv@virginia.edu

## ABSTRACT

The degree-corrected block model (DCBM), latent space model (LSM), and $\beta$-model are all popular network models. We combine their modeling ideas and propose the logit-DCBM as a new model. Similar as the $\beta$-model and LSM, the logit-DCBM contains nonlinear factors, where fitting the parameters is a challenging open problem. We resolve this problem by introducing a cancellation trick. We also propose R-SCORE as a recursive community detection algorithm, where in each iteration, we first use the idea above to update our parameter estimation, and then use the results to remove the nonlinear factors in the logit-DCBM so the renormalized model approximately satisfies a low-rank model, just like the DCBM. Our numerical study suggests that R-SCORE significantly improves over existing spectral approaches in many cases. Also, theoretically, we show that the Hamming error rate of R-SCORE is faster than that of SCORE in a specific sparse region, and is at least as fast outside this region.

## 1 INTRODUCTION

Community detection is a problem of major interest in network analysis (e.g., see (Goldenberg et al., 2010), a survey paper). Consider an undirected network with $n$ nodes and $K$ communities $\mathcal{C}_1, \mathcal{C}_2, \ldots, \mathcal{C}_K$ (a community is a group of nodes with similar behaviors). Let $A \in \mathbb{R}^{n,n}$ be the adjacency matrix, where $A_{ij} = 1$ if and only if there is an edge between node $i$ and $j$, $1 \leq i \neq j \leq n$. Conventionally, we do not count self edges, so $A_{ij} = 0$ if $i = j$. As in many works on community detection (e.g., Chen et al. (2018); Zhao et al. (2012); Yuan et al. (2022a)), we assume that each node belongs to exactly one of the $K$ communities. For each $1 \leq i \leq n$, we encode the community label of node $i$ by a $K$-dimensional vector $\pi_i$ (which is unknown to us) such that

$$\pi_i = e_k \quad \text{if and only if node } i \in \mathcal{C}_k \qquad (e_k: k\text{-th standard Euclidean basis vector of } \mathbb{R}^K). \quad (1)$$

The goal of community detection is to use $(A, K)$ to cluster all $n$ nodes into $K$ communities/groups.

The degree-corrected block model (DCBM) (Karrer & Newman, 2011) is a popular network model. Suppose we use a free parameter $\theta_i > 0$ to model the *degree heterogeneity* of node $i$, $1 \leq i \leq n$. For a non-negative matrix $P \in \mathbb{R}^{K,K}$, DCBM assumes that the upper triangular entries of $A$ are independent Bernoulli variables satisfying

$$\mathbb{P}(A_{ij} = 1) = \theta_i \theta_j \pi_i' P \pi_j \qquad \Longleftrightarrow \qquad \log(\mathbb{P}(A_{ij} = 1)) = \log(\theta_i) + \log(\theta_j) + \pi_i' Q \pi_j, \quad (2)$$

where $Q$ is a $K \times K$ matrix such that $Q = \log(P)$ entry-wise. When all $\theta_i$ are equal, DCBM reduces to the well-known *Stochastic Block Model (SBM)* (Holland et al., 1983). Note that as $0 \leq \mathbb{P}(A_{ij} = 1) \leq 1$, so implicitly, DCBM has imposed a set of constraints on its parameters:

$$\log(\theta_i) + \log(\theta_j) + \pi_i' Q \pi_j \leq 0, \qquad 1 \leq i, j \leq n. \quad (3)$$

These constraints make an already complicated setting even more complicated, so we desire to remove them if possible. Also, if the matrix $Q$ is positive definite, then $Q = U'U$ for a matrix $U \in \mathbb{R}^{K,K}$. In this special case, we can rewrite (2) as

$$\log(\mathbb{P}(A_{ij} = 1)) = \log(\theta_i) + \log(\theta_j) + z_i' z_j, \qquad \text{where } z_i = U \pi_i, 1 \leq i \leq n. \quad (4)$$

The latent space model (LSM) (Hoff, 2005) and the $\beta$-model (Chatterjee et al., 2011) are also popular network models. Denote the logit function by $\text{logit}(x) = \log(x/[1-x])$, $0 < x < 1$. In a representative form, for so-called latent positions $z_1, \ldots, z_n \in \mathbb{R}^K$, the LSM assumes

$$\text{logit}(\mathbb{P}(A_{ij} = 1)) = \log(\theta_i) + \log(\theta_j) + z_i' z_j. \tag{5}$$

Compared (5) with (4), the only difference is the log-link is replaced by the logit-link, so at least in the special case where $Q$ is positive definite, two models are similar. Also, if we drop the $z_i' z_j$ term on the RHS, then (5) reduces to the $\beta$ model (where we only have one community, i.e., $K = 1$).

However, despite the similarity, to many statisticians, (5) is highly preferred. The main reason is that, for binary data, the model recommended by textbooks is the logistic regression model (e.g., (Hastie et al., 2009, Section 4.4) and Dobson & Barnett (2018)), where the logit-link function was argued to be the most natural. Additionally, some popular Python packages, such as scikit-learn, frequently use the logit-link function. In fact, in the LSM case, since $\text{logit}(\mathbb{P}(A_{ij} = 1))$ can take any values in $(-\infty, \infty)$, we do not have the constraints (see (3)) as the DCBM case.

To combine the modeling ideas of all three models, we propose the *logit-DCBM*, where we assume

$$\text{logit}(\mathbb{P}(A_{ij} = 1)) = \log(\theta_i) + \log(\theta_j) + \pi_i' Q \pi_j, \quad \text{with } Q = \log(P) \text{ entrywise as above.} \tag{6}$$

Since we use the logit-link function, we do not need the constraints (3) as in the DCBM case. Also, we can view (6) as an extension of (2). Moreover, since we do not require $Q$ to be positive definite in (6), so (6) also extends (5) to a broader setting. Last, 6) reduces to the $\beta$-model if we let $Q = 0$.

In summary, we propose the logit-DCBM as a nonlinear variant of DCBM so hopefully it is more broadly acceptable, especially for researchers with a strong preferences in nonlinear network models (such as the LSM) and in using logistic regression type model for binary data.

We now rewrite the logit-DCBM in the matrix form. Note that under the model, $\mathbb{P}(A_{ij} = 1) = N_{ij} \cdot \theta_i \theta_j \pi_i' P \pi_j$, where $N_{ij} = [1 + \theta_i \theta_j \pi_i' P \pi_j]^{-1}$ is a nonlinear term. Let $N = (N_{ij})$, $\Theta = \text{diag}(\theta_1, \ldots, \theta_n) \in \mathbb{R}^{n,n}$ and $\Pi = [\pi_1, \ldots, \pi_n]'$. For any matrix $\Omega \in \mathbb{R}^{n,n}$, let $\text{diag}(\Omega) \in \mathbb{R}^{n,n}$ be the diagonal matrix where the $k$-th diagonal entry is $\Omega_{kk}$. Let $W \in \mathbb{R}^{n,n}$ be the matrix where $W_{ij} = A_{ij} - \mathbb{E}[A_{ij}]$ if $i \neq j$ and $W_{ij} = 0$ otherwise. Let $\circ$ denote the Hadamard (or entry-wise) product (Horn & Johnson, 1985). Under the logit-DCBM model (6),

$$A = \Omega - \text{diag}(\Omega) + W, \qquad \text{with } \Omega = N \circ \widetilde{\Omega} \quad \text{and} \quad \widetilde{\Omega} = \Theta \Pi P \Pi' \Theta. \tag{7}$$

Note that $\text{rank}(\widetilde{\Omega}) = K$, but due to the matrix of nonlinear factors $N$, $\text{rank}(\Omega)$ may be much larger than $K$. For this reason, (7) is not a low-rank model in general.

**Remark 1**. Since $N_{ij} \approx 1$ when $\theta_i \theta_j \pi_i' P \pi_j \approx 0$, one may think that the DCBM and logit-DCBM are close to each other. This is not true. First, $\theta_i \theta_j \pi_i' P \pi_j$ are not necessarily small for all $i, j$. Second, even if $\widetilde{\Omega}$ and $N \circ \widetilde{\Omega}$ are close in each entry, their spectra and norms can be very different.

**Literature review and our contribution**. The logit-DCBM (and all other models mentioned above) are so-called latent variable models, where $\Pi$ is the matrix of latent variables. For latent variable models, the EM algorithm (e.g., Dempster et al. (1977)) is a well-known approach. However, EM algorithm is computationally expensive, lacks of theoretical guarantee for high dimensional setting as we have here, and does not perform well when the networks are sparse. For network data, penalization approach is popular, and in the DCBM setting, there are many interesting works (e.g., Chen et al. (2018); Zhao et al. (2012)). However, since the DCBM is a latent variable model with many unknown parameters, these methods usually involve a non-convex optimization, where a good initialization is crucial. Also, penalization approaches are usually computationally relatively slow and hard to analyze. We can extend these approaches to LSM (Ma & Yuan, 2020) and logit-DCBM, but due to the nonlinearity in LSM and logit-DCBM, these issues persist.

For these reasons, spectral approaches for network data are especially appealing. Compared with EM algorithm and penalization approaches, spectral approaches are conceptually simpler, computational faster, and also easier (at least for the DCBM) to analyze. In the classical spectral approach, we cluster by applying $k$-means to the $n$ rows of the matrix $\widehat{\Xi} = [\hat{\xi}_1, \ldots, \hat{\xi}_K]$, where $\hat{\xi}_k$ is the $k$-th eigenvector of $A$. However, due to frequently observed phenomenon of *severe degree heterogeneity* in network data, such an approach frequently performs poorly. To fix the problem, (Jin, 2015) proposed SCORE as a new spectral approach. In the DCBM setting, SCORE was shown to have fast

convergence rates (e.g., Jin (2015); Jin et al. (2021b)). Also, in a survey paper (Ke & Jin, 2023), SCORE was compared with many algorithms on many real networks, where it was shown to be competitive in real data performances.

Motivated by these, we wish to extend SCORE to our setting. The challenge is, the success of SCORE critically depends on the fact that the DCBM is a low-rank model, but unfortunately, the logit-DCBM is not a low-rank model (see above).

We adapt SCORE to our setting by proposing the *Recursive-SCORE (R-SCORE)*: we initialize by a possibly crude estimate for $\Pi$ (denoted for $\widehat{\Pi}$), and then use $A$ and $\widehat{\Pi}$ to estimate $N$ (denoted by $\widehat{N}$). We then update $A$ by $A \oslash \widehat{N}$ ($\oslash$ denotes the entry-wise division) and repeat the above process for a number of times. The main idea here is that, if $\widehat{N} \approx N$, then $A \oslash \widehat{N} \approx A \oslash N = \widetilde{\Omega} - \mathrm{diag}(\widetilde{\Omega}) + W \oslash N$, where the RHS is a low-rank model (recall that $\mathrm{rank}(\widetilde{\Omega}) = K$).

The challenge is, how to estimate $N$ is a *difficult problem*, even if $\Pi$ is known. In fact, when $\Pi$ is known, we can restrict the network to each of the $K$ communities, where within each community, the logit-DCBM reduces to the $\beta$-model (which is a symmetrical version of the $p_1$ model (Holland & Leinhardt, 1981)). How to estimate $N$ in the $\beta$-model is a well-known open problem, as explained in the survey paper (Goldenberg et al., 2010) (see also Rinaldo et al. (2010)): "A major problem with the $p_1$ and related models, recognized by Holland and Leinhardt, is the lack of standard asymptotics, ..., we have no consistency in results for the maximum likelihood estimates".

We tackle this with a cancellation trick. Construct two types of cycles. For each type, the expected cycle count is a big sum of many terms, where due to the matrix of nonlinear factors $N$, we can not derive a simple expression. Fortunately, in the ratio of the two big sums, *the nonlinear factors in one big sum cancel with those in the other, and the ratio has a simple and closed-form expression.*

Therefore, if $\Pi$ is known, then the idea gives rises to a simple and convenient way to estimate $N$. Note that this also solves the open problem for the $\beta$-model aforementioned. In our case, $\Pi$ is unknown, but we can first obtain a possibly crude estimate $\widehat{\Pi}$, and then use $\widehat{\Pi}$ and the idea above to obtain an estimate $\widehat{N}$ for $N$. We can then repeat the two steps as in the R-SCORE.

**Remark 2**. As many recent procedures rely on a low-rank network model, the above idea is not only useful for adapting SCORE to our setting, but is also helpful in adapting other ideas (e.g., those on global testing (Jin et al., 2021a) and on estimating $K$ (Jin et al., 2023)) to our setting.

It remains to analyze SCORE and R-SCORE for the logit-DCBM model. Note that while the Hamming clustering error of SCORE was analyzed before (e.g., Jin (2015); Jin et al. (2021b)), but the focus were on the simpler DCBM model, where the analysis critically depends on that the DCBM is a low-rank model. Unfortunately, the logit-DCBM is not a low-rank model, so it is unclear how to extend the results in Jin (2015); Jin et al. (2021b) to our setting. Note also that R-SCORE is a new algorithm, and has never been analyzed before.

For any community detection procedures, we measure the performance by the Hamming clustering error. In the logit-DCBM model, we can always write

$$A = \widetilde{\Omega} + (N - \mathbf{1}_n \mathbf{1}_n') \circ \widetilde{\Omega} - \mathrm{diag}(\Omega) + W, \qquad \text{where } \widetilde{\Omega} \text{ is a low-rank matrix,}$$

and $(N - \mathbf{1}_n \mathbf{1}_n') \circ \widetilde{\Omega}$ can be viewed as a non-linear perturbation of $\widetilde{\Omega}$. We show that the Hamming error rate of SCORE is upper bounded by

$$C[\lambda_1(\widetilde{\Omega}) + \|(N - \mathbf{1}_n \mathbf{1}_n') \circ \widetilde{\Omega}\|]/\lambda_K^2(\widetilde{\Omega}).$$

This is the first time we derive a bound for the Hamming clustering error of SCORE for a nonlinear network model. Compared with existing works on DCBM (e.g., Jin et al. (2021a; 2023)), the analysis is quite different.

The Hamming error of R-SCORE is much harder to analyze, for many reasons. First, R-SCORE is a recursive algorithm, where the next step depends on the pervious one. Second, $\widehat{N}$ (the estimate for $N$) is a complicated nonlinear function of $A$, which depends not only on the clustering errors in the previous step, but also on the cycle count step aforementioned (where the analysis is non-standard).

Fortunately, we manage to derive an upper bound for the Hamming error rate of R-SCORE. To save space, we consider a special case here, leaving more general cases to Section 3.1. Consider the

special case where for all $1 \leq i \leq n$, $c_0 n^{-\beta} \leq \theta_i \leq c_1 n^{-\beta}$, where $\beta \in (0, 1/2)$ and $c_1 > c_0 > 0$ are constants. In this case, the Hamming error rates of SCORE are R-SCORE are upper bounded by $C n^{-a_0(\beta)}$ and $n^{-a_1(\beta)}$, respectively, where

$$a_0(\beta) = \min\{(1 - 2\beta), 4\beta\}, \quad a_1(\beta) = \min\{(1 - 2\beta), 6\beta\}, \quad \text{and } a_1(\beta) > a_0(\beta) \text{ when } \beta < 1/6.$$

Therefore, when $0 < \beta < 1/6$, the rate of R-SCORE is faster than that of SCORE, and two rates are the same when $1/6 < \beta < 1/2$ (the interesting range of $\beta$ is $0 < \beta < 1/2$; when $\beta > 1/2$, the signal-noise ratio is so low that no procedure could succeed).

We have the following contributions: (a) propose the logit-DCBM as an extension of the LSM, $\beta$-model, and DCBM, and as a more appealing network model, (b) introduce a cancellation trick and use it to solve an open problem for the $\beta$-model, as well as for an open problem for the logit-DCBM in the idealized case where $\Pi$ is known, and (c) propose R-SCORE as recursive approach to community detection with the logit-DCBM, (d) for the first time, we derive upper bounds for the Hamming error rates of SCORE and R-SOCRE for the logit-DCBM (which is a nonlinear network model), and (e) show that the rate of R-SCORE is faster than that of SCORE in a specific sparse region, and is at least as fast outside the region.

In summary, we propose the logit-DCBM as a nonlinear variant of DCBM, so it will be more broadly accepted. The nonlinear factors make the logit-DCBM harder to fit, but with a cancellation trick, we can successfully convert the model back to DCBM approximately, so we can continue to enjoy all nice properties the DCBM has. We also propose R-SCORE as a fast spectral approach where the error rate is faster than that of applying SCORE directly to the logit-DCBM.

**Content and notation**. Section 2 introduces the cancellation trick. Section 3 introduces the R-SCORE algorithm and theoretical analysis. Section 4 contains some numerical study. Section 5 discusses connections to other problems. In this paper, $\circ$ and $\oslash$ denote the entry-wise product and division, respectively. For $1 \leq k \leq K$, we use $e_k$ to denote the $k$-th standard basis vector of $\mathbb{R}^K$. For any $n \geq 2$, $I_n$ denotes the $n \times n$ identity matrix and $\mathbf{1}_n \in \mathbb{R}^n$ denotes the vector of all ones. For any two sequence of non-negative numbers $\{a_n\}$ and $\{b_n\}$, we write $a_n \gg b_n$ if $b_n/a_n = o(1)$ (similar for $a_n \ll b_n$), and we write $a_n \asymp b_n$ if $c_0 b_n \leq a_n \leq c_1 b_n$ for some constants $c_1 > c_0 > 0$. We use $C$ to stand for a generic constant, which may vary from one occasion to another.

## 2 AN IDEA FOR CANCELLING NONLINEAR TERMS IN BIG SUMS

We introduce the cancellation trick by considering two seemingly new problems. Although it seems a digression from our original purposes, the two problems are interesting in their own right, and provide the foundation for the refitting step of R-SCORE below. Consider the first problem. Suppose we have a matrix $A \in \mathbb{R}^{n_1, n_2}$ with independent Bernoulli entries, where $\Omega_{ij} = \mathbb{P}(A_{ij} = 1) = x_0 N_{ij} \theta_i \theta_j$, $x_0 > 0$, $\theta_i > 0$, with $N_{ij} = [1 + x_0 \theta_i \theta_j]^{-1}$. Here, $\theta_i$ are known but $x_0$ is not, and the interest is to estimate $x_0$. We may estimate $x_0$ by the maximum likelihood estimate (MLE), but it does not have a closed form and may be computationally slow, so we desire a new approach.

**Lemma 2.1** *We have $(I) = x_0(II)$, where $(I) = \sum_{i,j} \Omega_{ij}$ and $(II) = \sum_{i,j} \theta_i \theta_j (1 - \Omega_{ij})$.*

**Proof.** As $(I) = x_0 \sum_{i,j} N_{ij} \theta_i \theta_j$ and $(II) = \sum_{i,j} N_{ij} \theta_i \theta_j$, the claim follows. □

The key is, due to the non-linear terms $N_{ij}$, it is hard to derive a closed-form formula for $(I)$ or $(II)$, but by our careful design, the ratio of $(I)/(II)$ has a very simple form. Now, to estimate $x_0$, let $\psi_n^{(1)} = \sum_{i=1}^{n_1} \sum_{j=1}^{n_2} A_{ij}$ and $\psi_n^{(2)} = \sum_{i=1}^{n_1} \sum_{j=1}^{n_2} \theta_i \theta_j (1 - A_{ij})$. By Lemma 2.1, $x_0 = \mathbb{E}[\psi_n^{(1)}]/\mathbb{E}[\psi_n^{(2)}]$, so a convenient estimate for $x_0$ is (note: the computational cost is $O(n_1 n_2)$):

$$\hat{x}_0 = \psi_n^{(1)}/\psi_n^{(2)} = [\sum_{i=1}^{n_1} \sum_{j=1}^{n_2} A_{ij}]/[\sum_{i=1}^{n_1} \sum_{j=1}^{n_2} \theta_i \theta_j (1 - A_{ij})]. \tag{8}$$

Consider the second problem. Suppose we have a network adjacency matrix $A \in \mathbb{R}^{n_1, n_2}$ satisfying the $\beta$-model. That is, the upper triangle of $A$ are independent Bernoulli satisfying $\Omega_{ij} \equiv \mathbb{P}(A_{ij} = 1) = N_{ij} \theta_i \theta_j$ where $N_{ij} = [1 + \theta_i \theta_j]^{-1}$, $1 \leq i \neq j \leq n_1$. The parameters $\theta_i > 0$ are unknown and the interest is to estimate them. Due to the nonlinear terms $N_{ij}$, the problem remains a difficult open problem in the literature, where classical approaches such as the MLE face grand challenges (e.g.,

Goldenberg et al. (2010); Karwa & Slavković (2016); Rinaldo et al. (2010)). We propose a new approach, motivated by the following lemma. For any $1 \le i \le n_1$, let $S_i = \{1, 2, \dots, n_1\} \setminus \{i\}$.

**Lemma 2.2** *Fix an odd number $m \ge 3$. We have (dist below stands for distinct)*

$$\frac{\sum_{i_2,\dots,i_m \in S_{i_1}(dist)} \Omega_{i_1 i_2}(1 - \Omega_{i_2 i_3}) \dots \Omega_{i_{m-2} i_{m-1}}(1 - \Omega_{i_{m-1} i_m})\Omega_{i_m i_1}}{\sum_{i_2,\dots,i_m \in S_{i_1}(dist)} (1 - \Omega_{i_1 i_2})\Omega_{i_2 i_3} \dots (1 - \Omega_{i_{m-2} i_{m-1}})\Omega_{i_{m-1} i_m}(1 - \Omega_{i_m i_1})} = \theta_{i_1}^2. \quad (9)$$

**Proof**. Note that

$$\Omega_{i_1 i_2}(1 - \Omega_{i_2 i_3}) \dots \Omega_{i_{m-2} i_{m-1}}(1 - \Omega_{i_{m-1} i_m})\Omega_{i_m i_1} = N_{i_1 i_2} N_{i_2 i_3} \dots N_{i_m i_1} \theta_{i_1}^2 \theta_{i_2} \dots \theta_{i_m}$$
$$(1 - \Omega_{i_1 i_2})\Omega_{i_2 i_3} \dots (1 - \Omega_{i_{m-2} i_{m-1}})\Omega_{i_{m-1} i_m}(1 - \Omega_{i_m i_1}) = N_{i_1 i_2} N_{i_2 i_3} \dots N_{i_m i_1} \theta_{i_2} \dots \theta_{i_m}.$$

Comparing the RHS, the only difference is the term $\theta_i^2$. Since on both the numerator and denominator of (9), the sum is only over $i_2, i_3, \dots, i_m$ with $i_1$ being fixed, the claim follows. □

Similarly, due to the non-linear terms $N_{ij}$, it is hard to derive a closed-form formula for both the numerator and denominator of (9), but by our design, the ratio in (9) has a very simple form. Let $\phi_{n,m}^{(1)}(i_1) = \sum_{i_2,\dots,i_m \in S_{i_1}(dist)} A_{i_1 i_2}(1 - A_{i_2 i_3}) \dots A_{i_{m-2} i_{m-1}}(1 - A_{i_{m-1} i_m})A_{i_m i_1}$ and $\phi_{n,m}^{(2)}(i_1) = \sum_{i_2,\dots,i_m \in S_{k,i_1}(dist)} (1 - A_{i_1 i_2})A_{i_2 i_3} \dots (1 - A_{i_{m-2} i_{m-1}})A_{i_{m-1} i_m}(1 - A_{i_m i_1})$. By Lemma 2.2,

$$\frac{\mathbb{E}[\phi_{n,m}^{(1)}(i_1)]}{\mathbb{E}[\phi_{n,m}^{(2)}(i_1)]} = \frac{\sum_{i_2,\dots,i_m \in S_{i_1}} \Omega_{i_1 i_2}(1 - \Omega_{i_2 i_3}) \dots \Omega_{i_{m-2} i_{m-1}}(1 - \Omega_{i_{m-1} i_m})\Omega_{i_m i_1}}{\sum_{i_2,\dots,i_m \in S_{i_1}} (1 - \Omega_{i_1 i_2})\Omega_{i_2 i_3} \dots (1 - \Omega_{i_{m-2} i_{m-1}})\Omega_{i_{m-1} i_m}(1 - \Omega_{i_m i_1})} = \theta_{i_1}^2.$$

Therefore, a reasonable estimator for $\theta_{i_1}$ is $\hat{\theta}_{i_1} = \sqrt{\phi_{n,m}^{(1)}(i_1)/\phi_{n,m}^{(2)}(i_1)}$. This solves the open problem aforementioned (see also Section 3). Especially, we may take $m = 3$ and estimate $\theta_i$ by

$$\hat{\theta}_i = \sqrt{\phi_{n,3}^{(1)}(i)/\phi_{n,3}^{(2)}(i)} = \sqrt{\frac{\sum_{j,k \in S_i, j \ne k} A_{ij}(1 - A_{jk})A_{ki}}{\sum_{j,k \in S_i, j \ne k} (1 - A_{ij})A_{jk}(1 - A_{ki})}}. \quad (10)$$

Alternatively, we may use a larger $m$, but the numerical performance is similar, while the analysis is much longer. For each fixed $m$, the computational cost is $O(n^2 d)$ (e.g., Jin et al. (2021a)), where $d$ is the maximum node degree.

**Remark 3**. Lemma 2.2 is readily extendable. For example, if there are positive functions $g$ and $h$ such that $g(\Omega_{ij}) = \theta_i \theta_j \pi_i' P \pi_j h(\Omega_{ij})$ for all $i, j$, then similarly

$$\frac{\sum_{i_2,\dots,i_m \in S_{i_1}(dist)} g(\Omega_{i_1 i_2})h(\Omega_{i_2 i_3}) \dots g(\Omega_{i_{m-2} i_{m-1}})h(\Omega_{i_{m-1} i_m})g(\Omega_{i_m i_1})}{\sum_{i_2,\dots,i_m \in S_{i_1}(dist)} h(\Omega_{i_1 i_2})g(\Omega_{i_2 i_3}) \dots h(\Omega_{i_{m-2} i_{m-1}})g(\Omega_{i_{m-1} i_m})h(\Omega_{i_m i_1})}) = \theta_{i_1}^2.$$

In summary, the two problems above (especially the second one) are difficult. In these problems, the quantities of interest are hidden in some big sums. Due to the nonlinear factor $N_{ij}$, it is hard to derive a closed-form formula for such big sums. However, if we can carefully construct two big sums, then we can cancel the nonlinear terms $N_{ij}$ by considering the ratio of the two big sums, and derive a closed-form formula for the quantity of interest. Such a cancellation trick gives rises to a convenient approach to solving the two problems above, and is readily extendable to many other settings (e.g., analysis of the p1 model for directed networks Holland & Leinhardt (1981), analysis of tensor and hyper-graphs Yuan et al. (2022b)).

Below in Section 3, we introduce R-SCORE as a recursive algorithm, where the ideas above play a key role in the refitting steps of R-SCORE. For space reasons, we defer the analysis of the above idea (i.e., $\hat{x}_0$ and $\hat{\theta}_i$) to the supplement; see Sections C.2-C.3 of the supplement for details.

## 3 COMMUNITY DETECTION BY R-SCORE FOR THE LOGIT-DCBM

We propose *Recursive-SCORE (R-SCORE)* for community detection, where the key is to use the ideas above in the refitting step; see Algorithm (1). The number of iteration is not critical, so we set $M = 10$ (R-SCORE typically converges in very few iterations). In each iteration, R-SCORE

consists a community detection step by SCORE (the SCORE step) and a refitting step. We choose SCORE for it is fast, competitive in real data analysis, and with fast error rates (e.g., Jin (2015); Jin et al. (2021b)), but we can also view our algorithm as a *generic algorithm*, where we can replace the SCORE by any other community detection approaches that are provably effective for DCBM.

We now discuss the SCORE step and refitting step of Algorithm 1 in detail. Consider the SCORE step (Jin, 2015) first. In this step, for an input matrix $A$ or $A \oslash \widehat{N}$, let $\hat{\xi}_1, \ldots, \hat{\xi}_K$ be the first $K$ eigenvectors, and let $\widehat{R} = [\hat{\xi}_2/\hat{\xi}_1, \ldots, \hat{\xi}_K/\hat{\xi}_1]$, where $\xi/\eta$ denotes the vector of entry-wise ratios. We cluster by applying the $k$-means to the $n$ rows of $\widehat{R}$, and let $\hat{\pi}_i$ be the estimated community label of node $i$. Let $\widehat{\Pi} = [\hat{\pi}_1, \ldots, \hat{\pi}_n]'$. Note that $\hat{\pi}_i$ takes values in $e_1, e_2, \ldots, e_K$ ($e_k$: $k$-th standard basis vector of $\mathbb{R}^K$).

---

**Algorithm 1** The Recursive SCORE (R-SCORE)

**Input**: $A$ and $K$. Initialize with an estimate $\widehat{\Pi}$ by SCORE. For $m = 1, 2, \ldots, M$,

- *Refitting*. Update $\widehat{N}$ using $A, \widehat{\Pi}$ in the most recent step, and the refitting step below.
- *SCORE*. Update $\widehat{\Pi}$ by applying SCORE to $A \oslash \widehat{N}$ with the most recent $\widehat{N}$.

**Output**: $\widehat{\Pi} = [\hat{\pi}_1, \ldots, \hat{\pi}_n]'$.

---

Consider the refitting step. Let $\widehat{\Pi} = [\hat{\pi}_1, \ldots, \hat{\pi}_n]'$ be the estimated $\Pi$ in the current iteration. Recall that even in the idealized case of $\widehat{\Pi} = \Pi$, refitting (i.e., how to estimate $N$) is a difficult and open problem. We tackle this with the idea in Section 2. In detail, let $\mathcal{C}_k = \{1 \leq i \leq n : \hat{\pi}_i = e_k\}$ be the $k$-th estimated community, $1 \leq k \leq K$, and let $\hat{S}_{k,i} = \widehat{\mathcal{C}}_k \setminus \{i\}$. By Lemma 2.2 and especially (10), we propose to estimate $\theta_i$ by

$$\hat{\theta}_i = \sqrt{\left( \sum_{j \neq k \in \hat{S}_{k,i}} A_{ij}(1 - A_{jk})A_{ki} \right) / \left( \sum_{j \neq k \in \hat{S}_{k,i}} (1 - A_{ij})A_{jk}(1 - A_{ki}) \right)}, \qquad \text{if } i \in \widehat{\mathcal{C}}_k. \quad (11)$$

This corresponds to the case of $m = 3$ of our idea in Section 2, but we can also use a larger $m$. Also, inspired by Lemma 2.1, we can estimate the matrix $P$ by

$$\widehat{P}_{k\ell} = \left[ \sum_{i \in \widehat{\mathcal{C}}_k} \sum_{j \in \widehat{\mathcal{C}}_\ell} A_{ij} \right] / \left[ \sum_{i \in \widehat{\mathcal{C}}_k} \sum_{j \in \widehat{\mathcal{C}}_\ell} \hat{\theta}_i \hat{\theta}_j (1 - A_{ij}) \right], \qquad 1 \leq k, \ell \leq K. \quad (12)$$

To appreciate the idea, consider the sub-matrix of $A$ by restricting the rows and columns to $\widehat{\mathcal{C}}_k$ and $\widehat{\mathcal{C}}_\ell$. In the idealized case where $\hat{\theta}_i = \theta_i$ and $\widehat{\mathcal{C}}_k = \mathcal{C}_k$, $1 \leq i \leq n$, $1 \leq k \leq K$, the mean of the sub-matrix satisfies the condition of Lemma 2.1 of Section 2. This gives rises to the estimates above. Finally, we update our estimate of $N$ by letting $\widehat{N}_{ij} = (1 + \hat{\theta}_i \hat{\theta}_j \widehat{P}_{k\ell})^{-1}$ if $i \in \widehat{\mathcal{C}}_k$ and $j \in \widehat{\mathcal{C}}_\ell$.

Note that by the discussion in Section 2, the computational cost of the refitting step is no more than $O(n^2 d)$, where $d$ is the maximum node degree. As a result, the computational cost of R-SCORE is no more than $O(n^2 d)$.

**Remark 4**. One may want to replace the refitting step by a simpler step, but how to do so remains unclear. Recall that even when $\Pi$ is known, how to estimate $(\Theta, P)$ is a challenging problem (e.g., Goldenberg et al. (2010); Karwa & Slavković (2016); Rinaldo et al. (2010)).

### 3.1 THEORETICAL RESULTS

For any community detection procedure, let $\widehat{\Pi}$ be the resultant estimate for $\Pi$, where each row of $\widehat{\Pi}$ takes values in $\{e_1, e_2, \ldots, e_K\}$. We measure the performance by the Hamming error rate:

$$r_n(\widehat{\Pi}) = \frac{1}{n} \min_{\mathcal{P}} \sum_{i=1}^{n} 1\{\hat{\pi}_i \neq \mathcal{P}\pi_i\}, \qquad \text{where } \mathcal{P} \text{ is any permutation in } \{1, 2, \ldots, K\}.$$

Let $\widehat{\Pi}^{score}$ and $\widehat{\Pi}^{rscore}$ be the $\widehat{\Pi}$ for applying SCORE and R-SCORE to the adjacency matrix $A$.

Note that under the logit-DCBM model,

$$A = \Omega - \text{diag}(\Omega) + W, \text{ where } \Omega = N \circ \widetilde{\Omega}, \ \widetilde{\Omega} = \Theta \Pi P \Pi' \Theta \text{ and } P \text{ has unit diagonal entries.} \quad (13)$$

The last item is a well-known *identifiability condition* (Jin et al., 2023). Since in most real networks, $K$ is relatively small, so we suppose that $K$ is fixed (this is only for technical simplicity and can be relaxed). Let $n_k$ be the number of nodes in the $k$-th community, $1 \leq k \leq K$. We assume

$$\min_k\{n_k\} \geq c_0 n, \quad \text{for some constant } 0 < c_0 \leq 1/K. \tag{14}$$

This is a frequently used and *mild balance condition among the $K$ communities* (e.g., Jin (2015); Jin et al. (2021a)). Also, we assume that there exists constants $c_2 \geq c_1 > 0$ such that

$$\bar{\theta} \to 0, \quad \text{and} \quad c_1\bar{\theta} \leq \theta_i \leq c_2\bar{\theta} \text{ for all } 1 \leq i \leq n, \quad \text{where } \bar{\theta} = \sum_{i=1}^{n} \theta_i/n. \tag{15}$$

This condition is also only for technical reasons, and can be largely relaxed. Furthermore, we assume there exists an constant $c_3 > 0$, such that

$$\sqrt{n}\bar{\theta} \cdot |\lambda_{\min}(P)| \geq c_3 \log(n), \quad \lambda_{min}(P): \text{smallest eigenvalue of } P \text{ in magnitude.} \tag{16}$$

In the special case where $A$ satisfies a DCBM, $\Omega = \widetilde{\Omega}$, and SCORE was analyzed before (e.g., Jin (2015); Jin et al. (2021b)), where it is known that the signal-noise ratio (SNR) is given by $|\lambda_K(\widetilde{\Omega})|/\lambda_1^{1/2}(\widetilde{\Omega})$ ($|\lambda_K(\widetilde{\Omega})|$ and $\lambda_1^{1/2}(\widetilde{\Omega})$ represent the signal and noise level respectively). In order for the Hamming error rate of SCORE tends to 0, it is necessary that the SNR $\to \infty$. Condition (16) is necessary for otherwise the SNR may tend to 0. Also, here $\lambda_{\min}(P)$ measures community dissimilarity. In the special case of $P = b\mathbf{1}_K\mathbf{1}_K + (1-b)I_K$, $0 < b < 1$, $\lambda_{\min}(P) = 1 - b$. Therefore, if $\lambda_{\min}(P) \to 0$, then $b \to 1$, and all $K$ communities are very similar. Condition (16) defines a class of *weak signal settings* where the problem of community detection is challenging. Lastly, consider $P\Pi'\Theta^2\Pi$. Let $\eta$ be the first right eigenvector of $P\Pi'\Theta^2\Pi$, we assume that $\eta$ is a positive vector and

$$\lambda_1(P\Pi'\Theta^2\Pi) - |\lambda_2(P\Pi'\Theta^2\Pi)| \geq c_4\lambda_1(P\Pi'\Theta^2\Pi), \quad \max_i \eta(i)/\min_i \eta(i) \leq c \tag{17}$$

This condition is necessary to guarantee that the first eigenvector is well-separated from the others and the SCORE normalization by the first eigenvector is well-defined, since the reciprocal of each entry of the first eigenvector cannot blow up. It is a mild condition by Perron's theorem on non-negative matrices. Similar condition can be found in Jin et al. (2023).

Note that while SCORE was analyzed before for the DCBM, it was not analyzed for the logit-DCBM, where the analysis is expected to be much harder. In the logit-DCBM, we have $\Omega = \widetilde{\Omega} + (N - \mathbf{1}\mathbf{1}') \circ \widetilde{\Omega}$. To avoid that the nonlinearity completely ruins the low-rank structure, we need

$$\|(N - \mathbf{1}_n\mathbf{1}_n') \circ \widetilde{\Omega})\|/|\lambda_K(\widetilde{\Omega})| \to 0; \tag{18}$$

Recall that SNR $= |\lambda_K(\widetilde{\Omega})|/\lambda_1^{1/2}(\widetilde{\Omega})$. The following theorem is proved in the supplement.

**Theorem 3.1** *Let $\widehat{\Pi}^{score}$ be the resultant estimate for $\Pi$ when we apply SCORE directly to $A$ and suppose (14)-(17) and (18) hold. With probability $1 - o(n^{-3})$,*

$$r_n(\widehat{\Pi}^{score}) \leq C[\|(N - \mathbf{1}_n\mathbf{1}_n') \circ \widetilde{\Omega}\|^2 + \lambda_1(\widetilde{\Omega})]/\lambda_K^2(\widetilde{\Omega}).$$

*In the special case where $A$ satisfies the DCBM, $N = \mathbf{1}_n\mathbf{1}_n'$, and $r_n(\widehat{\Pi}^{score}) \leq C\lambda_1(\widetilde{\Omega})/\lambda_K^2(\widetilde{\Omega})$.*

Next, we consider R-SCORE. Since R-SCORE is a recursive algorithm, it is useful to present a result that is *applicable in general cases*. Consider an estimate for $\widetilde{\Omega}$ in the form of $\widehat{\widetilde{\Omega}} = \widehat{\Theta}\widehat{\Pi}\widehat{P}\widehat{\Pi}'\widehat{\Theta}$. By our construction, $\widehat{N}_{ij} = 1/(1 + \widehat{\widetilde{\Omega}}_{ij})$. Suppose that with probability $1 - o(n^{-3})$,

$$\|\widehat{P} - P\|_{\max} \ll \min\{1, |\lambda_{\min}(P)|\bar{\theta}^{-1}\}, \quad \|\widehat{\Pi} - \Pi\|(\sqrt{n}\,|\lambda_{\min}(P)|)^{-1}\bar{\theta} \to 0, \tag{19}$$

and

$$\|(N \oslash \widehat{N} - \mathbf{1}_n\mathbf{1}_n') \circ \widetilde{\Omega}\| = o(|\lambda_K(\widetilde{\Omega})|), \quad \|(N \oslash \widehat{N} - \mathbf{1}_n\mathbf{1}_n')\|_F = o(\lambda_1(\widetilde{\Omega})). \tag{20}$$

The following lemma is proved in the supplement.

**Lemma 3.1** *Suppose (14)-(17) hold. Let $\widehat{\Pi}$ be the result of applying SCORE to $A \oslash \widehat{N}$ where (19)-(20) and $\hat{\theta}_i < C\bar{\theta}$ hold. With probability $1 - o(n^{-3})$,*

$$r_n(\widehat{\Pi}) \leq \frac{C[\|(N \oslash \widehat{N} - \mathbf{1}_n\mathbf{1}_n') \circ \widetilde{\Omega}\|^2 + \tau_n^2 + \lambda_1(\widetilde{\Omega})]}{\lambda_K^2(\widetilde{\Omega})}, \text{where } \tau_n = \sqrt{n}\bar{\theta}^3[\sqrt{n}\|\widehat{P} - P\|_{\max} + \|\widehat{\Pi} - \Pi\|].$$

We now show that (19)-(20) hold in many settings. Our numerical study shows that R-SCORE typically converges in just one iteration, so for convenience in analysis, we consider R-SCORE with one iteration from now on in this section. Write for short $\delta_n = \max\{\|(N - \mathbf{1}_n\mathbf{1}_n') \circ \widetilde{\Omega}\|^2, \lambda_1(\widetilde{\Omega})\}/\lambda_K^2(\widetilde{\Omega})$. Theorem 3.2 is proved in the supplement.

**Theorem 3.2** *Suppose (14)-(17) hold and* $\delta_n/\min\{\bar{\theta}^2, \bar{\theta}|\lambda_{\min}(P)|, |\lambda_{\min}(P)|^2/\bar{\theta}^2\} \to 0$. *Let* $\widehat{\Pi}^{rscore}$ *be the estimate for* $\Pi$ *by applying R-SCORE to A, and let* $(\widehat{\Theta}, \widehat{P}, \widehat{\Omega}, \widehat{N})$ *be the corresponding estimates for* $(\Theta, P, \Omega, N)$ *in the refitting step of R-SCORE. We have that (19)-(20) hold and that with probability* $1 - o(n^{-3})$,

$$r_n(\widehat{\Pi}^{rscore}) \leq \frac{C}{\lambda_K^2(\widetilde{\Omega})}\Big(\lambda_1(\widetilde{\Omega}) + n\bar{\theta}^4\log(n) + n^2\bar{\theta}^2\delta_n^2 + n^2\bar{\theta}^6\delta_n\Big).$$

**Corollary 3.1** *Suppose (14)-(17) hold,* $|\lambda_{\min}(P)| \geq C$ *for a constant* $C > 0$, *and* $n\bar{\theta}^4 \to \infty$. *Let* $\widehat{\Pi}^{score}$ *and* $\widehat{\Pi}^{rscore}$ *be the estimates for* $\Pi$ *by applying SCORE and R-SCORE to the adjacency matrix A, respectively. With probability* $1 - o(n^{-3})$,

$$r_n(\widehat{\Pi}^{score}) \leq C\Big(\frac{1}{n\bar{\theta}^2} + \bar{\theta}^4\Big), \qquad r_n(\widehat{\Pi}^{rscore}) \leq C\Big(\frac{1}{n\bar{\theta}^2} + \bar{\theta}^6 + \frac{\log(n)}{n}\Big).$$

With a more careful analysis, we conjecture that the condition of $n\bar{\theta}^4$ can be removed, and the rate of R-SCORE is at least as fast as that of SCORE in the whole range of interest (note that the proof of Theorem 3.2 and Corollary 3.1 is already hard and relatively long). If we calibrate $\bar{\theta} = n^{-\beta}$ for a constant $\beta > 0$, then in order for the SNR $\to \infty$ (e.g., see (18)), we must have $0 < \beta < 1/2$. In this range, $r_n(\Pi^{score}) \leq Cn^{-a_0(\beta)}$ and $r_n(\Pi^{rscore}) \leq Cn^{-a_1(\beta)}$, where

$$a_0(\beta) = \left\{ \begin{array}{ll} 4\beta, & 0 < \beta \leq 1/6, \\ 1 - 2\beta, & 1/6 < \beta < 1/2, \end{array} \right. \qquad a_1(\beta) = \left\{ \begin{array}{ll} 6\beta, & 0 < \beta \leq 1/8, \\ (1 - 2\beta), & 1/8 < \beta \leq 1/6, \\ (1 - 2\beta), & 1/6 < \beta < 1/2. \end{array} \right. ;$$

see Figure 1. Therefore, when $0 < \beta < 1/6$, the Hamming error rate of R-SCORE is faster than that of SCORE. When $\beta > 1/6$, such a conclusion may also be true, as the current bound may be conservative: the Hamming error rate for R-SCORE depends on a complicated data dependent matrix $\widehat{N}$, and the error bound can hopefully be improved with more careful analysis.

**Remark 5.** The proof of Theorem 3.2 and Corollary 3.1 is hard, for $\widehat{N}$ has a complicated form: recall that $\widehat{N}_{ij} = 1/(1 + \widehat{\widetilde{\Omega}}_{ij})$ and $\widehat{\widetilde{\Omega}} = \widehat{\Theta}\widehat{\Pi}\widehat{P}\widehat{\Pi}'\widehat{\Theta}$, where $\widehat{\Pi}$ is the SCORE estimate for $\Pi$, and $(\widehat{\Theta}, \widehat{P})$ are constructed using $\widehat{\Pi}$, $A$, and a cancellation trick. Note that even when $\Pi$ is known, how to estimate $(\Theta, P)$ is a nontrivial problem and we resolve it with a cancellation trick.

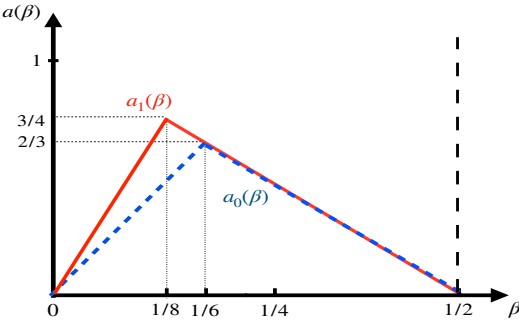

Figure 1: Comparison of error rates ($x$-axis: $\beta$. $y$-axis: $a_0(\beta)$ (blue) and $a_1(\beta)$ (red)) .

## 4 SIMULATION RESULTS

We compare R-SCORE with SCORE and a non-convex penalization MLE-based approach by (Ma et al., 2020), which we refer to as npMLE. We compare with npMLE for (Ma et al., 2020) deals with

the LSM and is probably the closest related work to our paper. Our study contains 3 experiments. In Experiment 1, we compare R-SCORE with SCORE (which is viewed as a benchmark). In Experiment 2, we study how the error rates of R-SCORE and npMLE change across different iterations (both algorithms are recursive). In Experiment 3, we compare the errors of R-SCORE and npMLE.

**Experiment 1.** *R-SCORE vs. SCORE.* The networks are simulated as follows: fixing $(n, K)$, we first simulate an $n \times n$ matrix $\Omega$ as in the logit-DCBM model as follows. Let $P = \beta \mathbf{1}_K \mathbf{1}_K + (1 - \beta) I_K$ and $\Pi = [\pi_1, \pi_2, \ldots, \pi_n]'$, where $\pi_i = e_k$ for $n_0 = n/K$ different $i$ (recall that $e_k$ is the $k$-th Euclidean basis vector of $\mathbb{R}^K$). Moreover, fixing a parameter $b_n > 0$, we generate $\theta$ as follows. We first draw $\theta_1^0, \theta_2^0, \ldots, \theta_n^0$ i.i.d. from a fixed distribution $F(\cdot)$, and then renormalize $\theta_0$ to get $\theta_i = b_n \cdot \theta_i / \|\theta\|$ for $1 \leq i \leq n$. Finally, we let $\widetilde{\Omega} = \Theta \Pi P \Pi' \Theta$ and $\Omega_{ij} = \widetilde{\Omega}_{ij} / (1 + \widetilde{\Omega}_{ij})$, $1 \leq i, j \leq n$. Once we have such a matrix $\Omega$, we use it to generate a binary adjacency matrix $A$.

In such settings, approximately, the Signal-to-Noise ratio (SNR) is $b_n(1 - \beta)$ (e.g., see Jin et al. (2021a)). It is desirable to choose settings that the SNR is neither too large or too small. Consider four settings (A), (B), (C) and (D). In Setting (A), we fix $(n, K) = (2400, 3)$ and $F = \text{Uniform}(0.01, 2)$. We choose $b_n = 60$ and $\beta = 23/30$ (and this way, SNR $= 14$). In Setting (B), we fix $(n, K) = (2500, 5)$ and $F = \text{Uniform}(0.1, 0.8)$. We choose $b_n = 70$ and $\beta = 0.65$ (and so SNR $= 24.5$). In Setting (C), we fix $(n, K) = (2400, 3)$ and $F = \text{Pareto}(10, 1)$. To avoid extremely severe degree heterogeneity, we truncate each $\theta_i^0$ at 200. We choose $b_n = 70$ and $\beta = 0.55$ (and this way, SNR $= 31.5$). In Setting (D), we fix $(n, K) = (2500, 5)$ and $F = \text{Pareto}(10, 1)$ with truncation at 100. We choose $b_n = 50$ and $\beta = 0.55$ (and so SNR $= 22.5$).

The results are in Figure 2, where the $x$-axis is the # of iterations $m$, and the $y$-axis is the corresponding error rate by R-SCORE (green dashed line: error rate for R-SCORE with $m = 0$, same as that of applying SCORE directly to the adjacency matrix $A$). In all settings above, the performance of directly applying SCORE ($m = 0$) is unsatisfactory. The improvements achieved by R-SCORE are significant, with substantially reduced error rates. This suggests that (a) the R-SCORE is successful as we expect, and (b) the iteration algorithm typically converges in very few iterations. Based on the numerical results, we believe that the refitting procedure steps in R-SCORE are effective: by re-normalization, they reduce the logit-DCBM model approximately to a low-rank model.

**Experiment 2.** *Error rates of R-SCORE and npMLE in different iterations.* Fix $(n, K) = (5400, 6)$. Let $\Pi$ be generated similarly to Experiment 1 except that $\pi_i = e_k$ for $n_k$ different $i$. Let $P = \begin{bmatrix} P_1 & P_2 \\ P_2 & P_1 \end{bmatrix}$, where $P_1 = 0.5\beta_1 \mathbf{1}_{K/2} \mathbf{1}_{K/2} + (1 - 0.5\beta_1) I_{K/2}$ and $P_2 = 0.5(\beta_1 + \beta_2) \mathbf{1}_{K/2} \mathbf{1}_{K/2}$.

We generate $\theta_i$ in the same manner as in Experiment 1 and similarly construct $\widetilde{\Omega}$ and $\Omega$ to generate a binary adjacency matrix $A$. In this experiment, we choose $(n_1, n_2, \cdots, n_K) = 200 \cdot (5, 1.5, 6, 3, 7.5, 4)$, $F = \text{Uniform}(0.01, 2)$, $b_n = 80$, and $(\beta_1, \beta_2) = (0.9, 0.6)$. Under this setting, the SNR is given by $b_n |\lambda_{\min}(P)| = 28$. For each $m = 1, 2, \ldots, 1000$, we apply R-SCORE and npMLE with $m$ iterations (SCORE is also included for comparison, which is the same as R-SCORE with $m = 0$). The result is in Figure 3 (left). We observed that (a) R-SCORE converges much more rapidly than the npMLE, and (b) the error rates of R-SCORE is significantly lower than that of SCORE, and is slightly lower than that of npMLE.

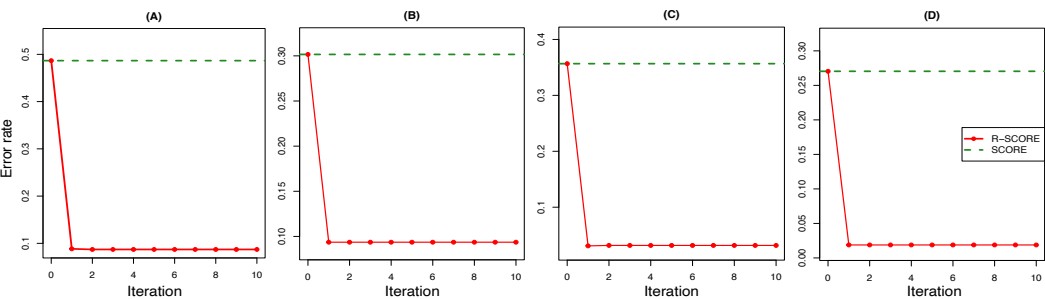

Figure 2: The error rates of R-SCORE for different $m$ (# of iterations). See Experiment 1 for setting details. SCORE is also included for comparision as a benchmark, which corresponding to R-SCORE with $m = 0$ ($x$-axis: $m$; $y$-axis: error rate).

**Experiment 3.** *R-SCORE vs. npMLE.* Consider the same setting as in Experiment 2, but we let $\beta_2$ vary: we set $b_n = 30$ and let $\beta_2$ range from $0.58$ to $0.7$ with a step size $0.02$ (other parameters remain the same). The SNR of the simulated network hinges on the smallest eigenvalue of $P$, which in turn hinges on $\beta_2$. The results (based on 20 repetitions) are in Figure 3 (right), which suggest that R-SCORE steadily outperforms npMLE for $\beta_2$ in the entire range. Also, R-SCORE is much faster than npMLE. In each repetition, it takes R-SCORE only 6 seconds, whereas it takes the npMLE more than 300 seconds (more than 50 times longer). This shows that R-SCORE not only is significantly faster than npMLE, but may also outperform the npMLE in many network settings.

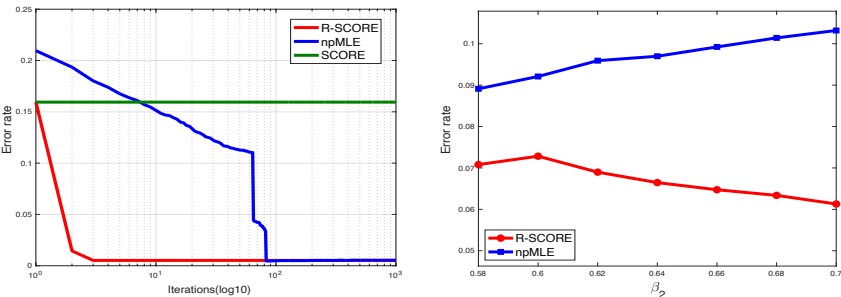

Figure 3: Left panel: The error rates by R-SCORE and npMLE for different $m$ (# of iterations). See Experiment 2 ($x$-axis: $\log_{10}(m)$, $y$-axis: error rates). SCORE is also included for comparison, which corresponds to R-SCORE with $m = 0$. Right panel: The error rates by R-SCORE and npMLE for different $\beta_2$. See Experiment 3 for setting details ($x$-axis: $\beta_2$; $y$-axis: error rate).

## 5 DISCUSSION

In this paper, we have made a three-fold contribution to the area of network community detection. First, we propose the logit-DCBM as a new network model. We argue that the logit-DCBM is more reasonable than the popular DCBM, but also poses a challenge. Second, to overcome the challenge, we propose a trick that can effectively cancel the effect of the nonlinear factors of the logit-DCBM model in some statistics (especially the ratio of two cycle-counts). Last, we propose R-SCORE as a new algorithm for community detection, and show that it can significantly improve over existing spectral approaches including the SCORE. Our idea is generalizable to many other settings. For example, the $p_1$ model by Holland and Leinhardt Holland & Leinhardt (1981) is one of the most popular models for directed networks with 1 community. Following the idea here, we can generalize it to a model with multiple communities, and extend R-SCORE for community detection with the new model. Also for example, the cancellation trick can be extended to many other settings (e.g., analysis of the $p_1$ model for directed networks Holland & Leinhardt (1981), text analysis Ke et al. (2023), tensor analysis Yuan et al. (2022b)) where the data matrix $A$ satisfies $\mathbb{E}[A] = N \circ \widetilde{\Omega}$ for a simple low-rank matrix $\widetilde{\Omega}$ and a matrix $N$ consisting nonlinear factors. Given that nonlinear models become increasingly more important in statistics and machine learning, the trick (and its extended form) may find increasingly more uses in many applications in the near future.

The cancellation trick is especially useful. In machine learning, we have many nonlinear latent variable models spreading in many areas (e.g., cancer clustering (Jin & Wang, 2016), text analysis (Ke & Wang, 2024), and empirical finance). Due to the nonlinearity, how to analyze such models is a challenging problem. In this paper, we propose an interesting cancellation trick using which we can effectively remove the nonlinear factor in some latent variable models. For space reasons, we only showcase this trick with a network setting, but the idea is extendable to other nonlinear latent space models. For this reason, our work may spark new research in many different directions in machine learning.

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
