# OpenReview forum: "Fitting Networks with a Cancellation Trick"
_ICLR.cc/2025/Conference — ICLR 2025 Poster_

### Official Review · Reviewer_CDCt · 2024-11-01

**Soundness:** 2
**Presentation:** 1
**Contribution:** 2
**Rating:** 5
**Confidence:** 3

**Summary:**

The paper proposes logit-DCBM (Degree Corrected Block Model) for community detection in graphs. Authors propose a method to overcome the presence of non-linear terms in the estimation problem. Authors refer to their method as the "cancelation trick". They propose an iterative method dubbed R-score (R for Recursive) which recursively applies the cancellation trick to the estimate. Authors provide theoretical results that guarantee the estimate is close to the true value.

**Strengths:**

Literature is well reviewed and contributions are well positioned with respect to existing art. Authors introduce their contributions with thorough explanations that are explained in easy-to-read language, despite the large amount of notation and math involved. Theoretical results are well presented and seem to indicate the merit of the method.

**Weaknesses:**

Presentation of this rather theoretical paper could benefit from a more formal statement of results with a Theorem which goes earlier in the paper and the body of the paper being devoted to explaining the intuition and implications of the result.

I do not find a clear indication of stopping criteria in the paper, neither did I find any discussion on computational complexity and scalability of the method.

Numerical experiments are not well presented. At the very least figures could have x and y labels.

**Questions:**

Numerical experiments do not show any benefit in iterating: after a single iteration the method seems to have converged. Can you explain this? was the test data too easy? did you try running your model on more challenging problems?

---

> ### Author Response · Authors · 2024-11-28
> **Reply to Reviewer CDCt**
>
> Thanks for a great summary.  We are very glad that the reviewer find our theoretical results are well presented and support the
> merit of our method. Below, we first provide clarifications on the weaknesses and then respond to the questions in detail.
> ***
> **Presentation style**: Thanks, but we have explained the theoretical results on Page 3-4, with explicit rates for the Hamming errors (see Line 161). It is unconventional to present a theorem in the introduction, especially for short papers like this one.
>
> ***
> **Stopping criteria and computational complexity**: The stopping criteria was mentioned in the first paragraph of Section 3 (Line 253-256).  The computation cost is specified in the paragraph right above Remark 4.
>
> ***
> **Numerical experiment**: Thanks and we have fixed this and revised the text.   We have also added two new experiments comparing R-SCORE with the non-convex penalization MLE-type algorithm (npMLE) by Ma, Ma, and Yuan (2020), which we refer it to as the npMLE.  See Section 4.
>
> ***
> **Response to question on numerical studies**: Thanks for the question. Regardless of the data settings, R-SCORE
> always converges very quickly ($\leq 5$ steps). For example, in the experiment reported last time, R-SCORE converges in $1$ step. In the newly added experiment, R-SCORE converges in $3$ steps.  The key idea of R-SCORE is to use a cancellation trick to estimate
> the nonlinear factors, and so to convert the logit-DCBM to a DCBM.  By design, such an algorithm usually converges quickly, for
> the estimate for each parameter in the nonlinear factor quickly become flat when we iterate. This is different from penalization approaches (e.g., Ma, Ma, Yuan (2020), which is also an MLE approach), where we need hundreds of iteration.   See Figure 3 (left)  in the supplement for how fast  two algorithms converge when we iterate.
>
> Note that a quick convergence is really not a disadvantage.  In the revision, we compare  R-SCORE with the MLE approach by Ma, Ma, Yuan (2020). In many settings, R-SCORE is not only much faster, but also has a smaller error rate in many settings. See for example Figure 3 (right).

---

### Official Review · Reviewer_urJ2 · 2024-11-03

**Soundness:** 3
**Presentation:** 2
**Contribution:** 1
**Rating:** 5
**Confidence:** 4

**Summary:**

The paper introduces the logit-DCBM as an extension of popular network models DCBM, LSM, and $\beta$-model. The authors propose a "cancellation trick" to tackle the nonlinearity challenges of the logit-DCBM, which allows the model to retain a form of low-rank approximation similar to DCBM. They also present an algorithm, R-SCORE, for community detection in networks based on the logit-DCBM. R-SCORE iteratively updates the model's parameter estimates, removing nonlinear factors and approximating a low-rank model for clustering. The paper shows that R-SCORE achieves a faster Hamming error rate in certain sparse regions than standard SCORE.

**Strengths:**

The logit-DCMB model provides a nice combination of the DCMB, LSM, and $\beta$ models.

The "cancellation trick" allows disregarding of nonlinear terms by means of a ratio of two large sums used as estimators. This is an interesting idea that can be applied to various other settings.

**Weaknesses:**

The scope of the paper is [quote] "to propose a nonlinear version of DCMB so that it will hopefully be more acceptable" and extend SCORE to it. I find this goal somewhat incremental and only marginally related to the ICLR community.

The paper is hard to follow, with many acronyms that are not spelt out and sentences that are ambiguous. An example: line 085, "The logit-DCBM (and all other models mentioned above) are so-called latent variable models, where Π is the matrix of latent variables. For latent variable models, the EM algorithm (e.g., Dempster et al. (1977)) is a well-known approach." What is EM? "approach" to do what?

**Questions:**

In eq (16), how can you guarantee |\lambda_{min}|>0?

In Thm 3.1, Lemma 3.1, Thm 3.2, Corollary 3.1: What is C? Are the C used the same across the three results?

In the simulations, can't you compare with additional baselines based on e.g. a MLE?

---

> ### Author Response · Authors · 2024-11-28
> **Reply to Reviewer urJ2**
>
> Thanks for the great summary and comments. We are happy to address the concerns.
> ***
> **The scope of the paper and relevance to ICLR community**: Thanks for the comment. First, we wish to point out that "proposing logit-DCBM" is only one of our contributions. We also have several other ones, including a novel cancellation trick, a new algorithm R-SCORE, and new theoretical results.  Especially, the cancellation trick is a new approach for removing nonlinearity in latent space models and can be generalized to other nonlinear links (please see our response to Reviewer 2iwW). An important message we convey in this paper is: MLE (which can only be solved by nonconvex optimization) is not the only option for attacking nonlinear problems. There exists a novel cancellation trick that removes nonlinearity and allows us to apply those power tools designed for linear settings (such as spectral approaches). We believe this is a significant contribution.
>
> Even this particular motivation of "studying logit-DCBM" is important by itself. Network modeling is an important problem in the  machine learning community, and what is the best model for real networks remains an open problem.  The DCBM, LSM, and
> $\beta$-model are all popular models, and  many works about these models have been published in machine learning conferences. Here are a few examples:
>
> - Neural Latent Space Model for Dynamic Networks and Temporal Knowledge Graphs. By Tony Gracious et al., AAAI 2021
> - Learning Guarantees for Graph Convolutional Networks in the Stochastic Block Model. By Wei Lu, ICLR 2022
>
> Our logit-DCBM is a broader model that extends the above models.  Therefore, our work is  an important step forward in network modeling.
>
> Second, our work is **closely related to machine learning**.   In machine learning,  we have many nonlinear latent variable models
> spreading in many areas,  including cancer clustering, empirical finance, text analysis, and network analysis. Due to the nonlinearity,
> how to analyze such models is a challenging problem. In this paper, we propose an interesting cancellation trick using which
> we can effectively remove the nonlinear factor in some latent variable models.
>
> Especially, we showcase this cancellation trick with a network community detection setting. The logit-DCBM is a nonlinear model, where how to do community detection is a challenging problem:  existing approaches are both computationally inefficient or hard to analyze.
> We show that, by removing the nonlinear factor in a logit-DCBM (a nonlinear network model), we can convert it approximately to a DCBM. We can then effectively apply (say) SCORE (a recent spectral approach) for community detection.
>
> For space reasons,  we only showcase this trick with a network setting, but the idea is extendable to other nonlinear latent space models. For this reason, our work may spark new research in many different directions in machine learning.
>
> ***
> **Hard to follow with many acronyms**: Thanks, but we believe our way for presenting the acronym is conventional: we present the full name when the acronym appears the first time, with references. For example, the full name of DCBM is presented in the first line of the abstract, and again in Line 36 of the introduction with a reference.  The full name of LSM is explained in Line 51 with a reference. The logit-DCBM is the direct combination of logit and DCBM, so  the is no need to present the full name.  For your comments on EM algorithm:  in most published works, people only cite it as the "EM algorithm", without explaining what EM stands for. The EM algorithm is a textbook algorithm (first proposed by Dempster et al (1977)) for estimating latent variable models.
> ***
> **Response to Question on $|\lambda_{\min}(P)|$**:  In the area of network community detection, it is conventional to assume $\mathrm{rank}(\Omega) = K$ (in the DCBM case)  or $\mathrm{rank}(\widetilde{\Omega}) = K$ (the logit-DCBM or LSM case).  Since $\mathrm{rank}(\widetilde{\Omega}) = K$ in our case,  we must have $\mathrm{rank}(P) = K$ by basic algebra.  Since $P$ is $K \times K$, $P$ is non-singular, and so $|\lambda_{min}(P)| > 0$.
> ***
> **Response to Question on $C$**: In this paper, $C$ stands for a generic constant which may vary from one occasion to another (such a use of notation is conventional in the literature). We have clarified this in the end of Section 1 (Content and notation).
> ***
> **Response to Question on simulations**: Thanks and this is a great comment.  It seems that the most appropriate MLE algorithm we can compare is the non-convex penalized MLE-based approach (npMLE) by Ma, Ma and Yuan 2020. The paper deals with the latent space model (LSM) and is probably the closest related work to our paper. We have added new two experiments (Experiment 2-3) in Section 4.  The experiments suggest
>
> - The R-SCORE runs much faster than the npMLE.
> - In many settings, R-SCORE has a lower error rates than the npMLE.

---

### Official Review · Reviewer_2iwW · 2024-11-04

**Soundness:** 4
**Presentation:** 4
**Contribution:** 4
**Rating:** 8
**Confidence:** 2

**Summary:**

This paper proposes R-SCORE (Recursive-SCORE), a generalization of the SCORE spectral algorithm to estimate the parameters and communities in a newly introduced "nonlinear" extension of the DCBM model [3] where a logit function is added to the probability. The R-SCORE algorithm alternatingly optimizes the estimates of the core matrix $\Pi$ and the offset terms $\theta$ with the SCORE algorithm and a custom refitting step respectively.  Theoretical results are shown both for the application of the original SCORE algorithm to the proposed extension logit-DCBM and to the application of the newly proposed R-DCBM algorithm





***References***

[1] Jin J. 2015 Annals of Stats.  "Fast community detection by SCORE"
[2] Jiashun Jin, Zheng Tracy Ke, Shengming Luo, and Minzhe Wang. "Optimal estimation of the number of network communities". Journal of the American Statistical association, 2023.

[3] Karrer, B. and Newman, M. E. J. (2011). Stochastic blockmodels and community structure in networks

**Strengths:**

1. Well-written
2. Appears rigorous and professional
3. A generalization of the DCBM model to the logit case is an interesting direction

**Weaknesses:**

Accessibility to readers less familiar with the literature on community detection

No table of notations

**Questions:**

Could you compare your rates with the classic rates for the simpler SBM model of Abbe et al. when the thetas are equal?

---

> ### Author Response · Authors · 2024-11-28
> **Reply to Reviewer 2iwW**
>
> We thank the reviewer for a great summary and we are very glad that the reviewer thinks our study depicts an optimistic view of the problems. Below we provide some clarifications on the weaknesses and address the question.
> ***
> **Accessibility**: We thank you for a great comment. Community detection is probably the most studied problem in network analysis, and we have revised the paper to make it more accessible to such readers.
>
> ***
> **Table of notations**: We do have a subsection called "Content and notation" in the end of Section 1. The table covers all notations we think that worthy of a clarification.
> ***
> **Response to Question**: Thanks for raising this great question, we mention two points.
> First, in the simpler SBM case, all $\theta_i$ are the same, so the logit-SBM model and the SBM are equivalent.  This is because the nonlinear matrix $N$ satisfies
> $$
> N_{ij} = \frac{e^{\theta_i + \theta_j}}{1 + e^{\theta_i + \theta_j}} = c_0,  \qquad \mbox{for a constant $c_0$, if all $\theta_i$ are the same}.
> $$
> Therefore, in this special case, the logit-DCBM reduces to an SBM (which is a special DCBM). Second, for DCBM, many algorithms have exponential error rates.  However, for general logit-DCBM,  due to the nonlinear factors, the optimal rates remain unknown, but it is likely that we only have polynomial rate. For example, suppose all $\theta_i$ are at the same order. Then the error rate of SCORE is
> $$
> \leq C e^{-C  n \bar{\theta}},
> $$
> which is exponential (e.g., Jin, Ke, Luo (2021)). For the logit-DCBM,
> we only be able to show that that the error rate of a procedure is
> $$
> \leq C (n \bar{\theta})^{-\alpha}, \qquad \mbox{for some $\alpha > 0$},
> $$
> which is polynomial.

---

### Official Review · Reviewer_wiFe · 2024-11-05

**Soundness:** 3
**Presentation:** 2
**Contribution:** 2
**Rating:** 5
**Confidence:** 3

**Summary:**

The authors first propose the logit-DCBM model as a nonlinear variant of the degree-corrected block model (DCBM), which essentially replaces the log link function with the logit link function. Replacing the link from log brings some nice properties to the model, such as removing some constraints on latent parameters. However, at the cost of introducing non-linearity and the parameter estimation becomes difficult.

The main technique is the cancellation trick, used to estimate nonlinear terms in the logit-DCBM, which simplifies parameter estimation and the authors claim that this addresses an open problem in the $\beta$-model parameter estimation (another one community variant which is the special case of their logit-DCBM). Additionally, the paper introduces R-SCORE for community detection, a recursive adaptation of the SCORE algorithm for the logit-DCBM. Finally, they derive upper bounds on the classification error rates for SCORE and R-SCORE for their proposed model logit-DCBM.

**Strengths:**

See summary.

**Weaknesses:**

Overall, I am not too sure why the logit-DCBM model is worth studying. The authors mention some motivation in lines 58-60 and lines 69-71, but I did not find it compelling enough. I found the motivation argued hand-heavily without any concrete and interesting reasons.
Line 58, ",to many statisticians, (5) is preferred.":  Which statisticians? References?
Line 59-60: I don't think the logistic regression is the only "recommended" model by text bool for binary data. Also, what does "recommending" even mean? Where does Hastie et al. 2009 recommend it?

See also questions. Questions 1 and 2 in particular, where I further raised my concern.

It is perhaps interesting to study logit-DCBM, but it is not clear to me right now based on what is written on the paper, and that's why I am currently leaning towards a borderline reject.

**Questions:**

1. I could not exactly make sense of the open question in the references given e.g. (Goldenberg et al., 2010). What is the open problem precisely? To analyze MLE? Or just to come up with an estimate? Or to come up with an estimate that satisfies certain properties? What is the precise technical open question?
2. What properties does your estimate have? As far as I can think, it is not unbiased, or is it? Is it consistent? What can you say further? For me, right now, it is just some estimate.
3. Ultimately, I really want to get a better understanding of Lemma 3 and how good this estimate is. I understood the statement and algebra used therein. But is there something more interesting? What is the effect of $m$ on the quality of the estimate? Does it not change anything in terms of quality? Or even if it does, we are looking at the statistics for $m=3$ for computational efficiency? Also, is there an intuitive way of thinking about what that statistics is capturing in (10) and its equivalent for higher $m$? And why I can expect it to be a reasonable estimate? Essentially, trying to get intuition on why cancellation happens.
4. Does the cancellation trick also apply to other link functions beyond logit? In either yes or no, could you please walk me through why for some standard link functions (e.g. probit or other relevant ones)?

Typos:
-In Lemma 2.1 Proof, is there an extra summation over $i$ again in $(I)$?
-In Line (222): ",the sum is over..",the index $i_2$ is repeated twice.

---

> ### Author Response · Authors · 2024-11-28
> **Reply to Reviewer wiFe (1)**
>
> Thanks for the summary and great comments. We are happy to address the concerns.
> ***
> **Response to Weaknesses**: We wish to re-iterate the motivation of our work, which is three-fold:
>
> * We introduce a cancellation trick, which can be useful for many nonlinear latent-variable models in statistics and machine learning. For reasons of space, we only use the logit-DCBM to showcase the trick, but the trick may be useful in much broader settings. We have explained this in ``Comments to all" above;  see details therein.
>
> * If we use DCBM, then we must impose a large number of constraints as in  equation (3). This is not desirable, especially if we want to fit the model using a penalization approach. If we use the logit-DCBM, then we do not need such constraints.
>
> * The logit-DCBM is an extension of the DCBM, LSM, and $\beta$-model, each of which is popular in network analysis. Using the logit-DCBM allows us to have a unified model, and thus helps reduce repetitive and overlap research in this area (and so save research time and efforts in the research community of network analysis).
>
> We believe we have explained the motivations carefully in the introduction, but we agree with you that more references could help. In this revision, we have added textbooks and popular packages there as reference, in all of which the logistic link is the default choice for handling binary data.  We also pointed out which place of Hastie et al (2009) recommends using logistic regressions.
>
> ***
> **Response to Question 1**: The open problem is all of these: "how to analyze MLE",  "how to come up with an estimate" and "how to come up with an estimate that satisfies certain property" (e.g., consistency).   In classical statistical settings, the MLE is usually consistent, and we can analyze the MLE with a standard approach (e.g., Lehmann and Casella, 2006, textbook).
> For our setting, standard analysis for MLE does not work, and there is no consistency results for the MLE.  It is therefore unclear (a) how to develop a new method, (b) what methods are consistent.  We have explained these in Line 118-120 carefully: "How to estimate N in the $\beta$-model is a well-known open problem, as explained in the survey paper (Goldenberg et al., 2010) (see also Rinaldo et al. (2010)): "A major problem with the $p_1$ and related models, recognized by Holland and Leinhardt, is the lack of standard asymptotics, ..., we have no consistency in results for the maximum likelihood estimates” (the $p_1$ model is a special case or ours).
>
> Our major contribution here is that,  using a cancellation trick we discover, we overcome the challenge by proposing an easy-to-use estimate and show that they are consistent, and lead to improved community detection results in the down-stream.
>
> ***
> **Response to Question 2**: The estimates are consistent with rigorous proofs.  In fact, under our settings,    $\mathbb{P}(\widehat{\Pi} \neq \Pi) = o(1)$, and the estimates for $P$ and $\theta_1, \ldots, \theta_n$ are all consistent.  Since that our ultimate goal is community detection (where we use such estimates in a middle step), so for reasons of space, we defer both the statements and proofs for consistency to the supplement: see Section C.2 and C.3 or Lemma C.1 and C.2 for details.  Second, R-SCORE is the combination of SCORE and these estimates. Our main results in Section 3.1 shows that R-SCORE improves SCORE on the error rate of community detection. This implies that these estimates behave as expected.
> ***

---

> ### Author Response · Authors · 2024-11-28
> **Reply to Reviewer wiFe (2)**
>
> **Response to Question 3**: Sorry but we do not have a Lemma 3 in our paper, but based on your description, we suppose you are referring to Lemma 2.2,  which is about using cancellation trick to estimate $\theta_i$.
>
> What we present in Section 2 is the so-called *oracle approach* in the literature,
> a frequently used idea for constructing new estimates.   The main idea of the oracle approach is,  we first find a way to reconstruct the parameters of interest in the  *idealized noiseless case*, we then mimic the idea in the noiseless case and propose an approach for the real case. The oracle approach has been proven to be very successful in the literature, many important modern ideas,  including classical PCA and the lasso (see "Uncertainty Principles and Signal Recovery"  by Donoho (1989)),  originated from the oracle approach.
>
> In our setting, the parameters of interest are $\theta_1, \theta_2, \ldots, \theta_n$, and the model is
> $$A = \Omega - \mathrm{diag}(\Omega) +W,
> $$
> where $W$ is the noise matrix, and $\mathrm{diag}(\Omega)$ only has a secondary effect. Therefore,
> to apply the oracle approach to our setting,
> - we first use $\Omega$ to construct $\theta_1, \ldots, \theta_n$ (this is the idealized noiseless case).
> This is done in Lemma 2.2, where we show that for each odd number $m \geq 3$,
> $$
> \theta_{i_1}^2 = \frac{\sum_{i_2, \ldots, i_m \in S_{i_1} (dist)}
> \Omega_{i_1 i_2} (1 - \Omega_{i_2 i_3}) \ldots \Omega_{i_{m-2} i_{m-1}} (1 - \Omega_{i_{m-1} i_m}) \Omega_{i_m i_1}
> }{\sum_{i_2, \ldots, i_m \in S_{i_1}(dist)} (1  -  \Omega_{i_1 i_2}) \Omega_{i_2 i_3}  \ldots (1 - \Omega_{i_{m-2} i_{m-1}}) \Omega_{i_{m-1} i_m} (1 -  \Omega_{i_m i_1})}.
> $$
> - we then mimic the idea and replace $\Omega$ by $A$ everywhere in the construction above (this gives rise to
> an estimate for the real case).  This is equation (10), where we set $m = 3$. For general $m$, the idea is similar, and for every $i_1$,  we estimate $\theta_i$ by
> $$
> \hat\theta_{i_1}^2 = \frac{\sum_{i_2, \ldots, i_m \in S_{i_1} (dist)}
> A_{i_1 i_2} (1 - A_{i_2 i_3}) \ldots A_{i_{m-2} i_{m-1}} (1 - A_{i_{m-1} i_m}) A_{i_m i_1}
> }{\sum_{i_2, \ldots, i_m \in S_{i_1}(dist)} (1  -  A_{i_1 i_2}) A_{i_2 i_3}  \ldots (1 - A_{i_{m-2} i_{m-1}}) A_{i_{m-1} i_m} (1 -  A_{i_m i_1})}.
> $$
> This is nothing but replacing all $\Omega$ by $A$ in equation (9) of Lemma 2.2.
>
> - we then justify the derived estimates $\hat{\theta}_{1}, \ldots, \hat{\theta}_n$
> are consistent for $\theta_1, \ldots, \theta_n$ rigorously.
> For space reasons, we present this in Lemma C.1-C.2 of the supplement.
> The idea why this works is similar to that of central limit theorem, though the proof is
> much more complicated.
>
> The above work for each fixed odd number $m \geq 3$.  But since the algorithm with $m = 3$ is already first-order optimal,  there is little reason to use a larger $m$ (for a larger $m$, we have similar numerical results, but the
> proof is much longer).
>
> ***
> **Response to Question 4**: Yes, the cancellation trick is applicable to much broader settings.
> In our setting, Lemma 2.2 holds because for all $i, j$ in the same community,
> $$
> \frac{\Omega_{ij}}{(1 - \Omega_{ij})}  = \theta_i \theta_j \pi_i' P \pi_j =   \theta_i \theta_j.
> $$
> We have a similar result if  there is a positive function $g$ and $h$ such that for all $i, j$ in the same community,
> $$
> \frac{g(\Omega_{ij})}{h(\Omega_{ij})}  = \theta_i \theta_j \pi_i' P \pi_j =  \theta_i \theta_j.
> $$
> In fact, by basic algebra, it is seen that for any odd number $m \geq 3$,
> $$
> \theta_{i_1}^2 = \frac{\sum_{i_2, \ldots, i_m \in S_{i_1} (dist)}
> g(\Omega_{i_1 i_2}) h(\Omega_{i_2 i_3}) \ldots  g(\Omega_{i_{m-2} i_{m-1}})  h(\Omega_{i_{m-1} i_m}) g(\Omega_{i_m i_1})
> }{\sum_{i_2, \ldots, i_m \in S_{i_1}(dist)}  h(\Omega_{i_1 i_2}) g(\Omega_{i_2 i_3})  \ldots h(\Omega_{i_{m-2} i_{m-1}}) g(\Omega_{i_{m-1} i_m})  h(\Omega_{i_m i_1})}.
> $$
> For revision, we have added a new remark, Remark 3, in Section 2.  See details therein.
>
> ***
> **Response to Typos**: Thank you and we have fixed them.

---

### Author Response · Authors · 2024-11-28
**Comments to All**

We thank all reviewers  for their valuable time and thoughtful comments. The reviewers have two main points or recommendations: (1) to clarify the relevance to machine learning, and (2) to add a numerical comparison with the MLE approach.

***
**For the first main point**: (a)  Our paper introduces a cancellation trick by which we can conveniently reduce a nonlinear latent variable setting to a linear latent variable setting, (b) network analysis is an important area in machine learning, with increasingly more interest,  and how to model and analyze social networks remains a largely open problem.

For (a): Compared with classical multivariate statistical analysis approaches, a major advantage of  machine learning approaches lies in modern ideas for tackling nonlinearity. However, for unsupervised learning  (e.g., clustering and fitting latent variable models),
how to tackle nonlinearity is a very challenging problems. Most existing machine learning approaches use non-convex optimization, with substantial efforts devoted to speed up algorithms, but still, many of non-convex optimization approaches remain slow computationally.

In this paper, we discover a simple cancellation trick that is able to remove nonlinearity effectively and so allows the users to take advantage of existing powerful methods for linear problems (e.g.,  PCA) and thus significantly reduce computational costs. This points out a new possible solution for tackling nonlinearity.

Since nonlinear latent variable models exist in many machine learning problems and have broad applications (e.g., in cancer clustering,  empirical finance, text analysis, and network analysis), we believe our paper is very relevant to the machine learning community.

Especially, the logit-DCBM is a nonlinear latent variable model, where how to do community detection is a challenging
problem:  existing approaches are either computationally inefficient, or hard to analyze, or both. We show that, by removing the nonlinear factor in the logit-DCBM (a nonlinear network model), we can convert it approximately to a DCBM. We can then effectively apply (say) SCORE (a recent spectral approach) for community detection. For networks with a few thousand nodes, this can speed up the computation by 50 times compared to using MLE.


For space reasons,  we only showcase this trick with a network setting, but the idea is extendable to other nonlinear latent space models. For this reason, our work may spark new research in many different directions in machine learning.  In the revised paper, we have added a paragraph in the end of Section 5, reflecting the points above. See details therein.

For (b):  Network analysis  is an important area  in machine learning. Many papers on network modeling and analysis haven been accepted at top machine learning conferences. For instance,

1) Effects of Graph Convolutions in Multi-layer Networks. By Aseem Baranwal et al.,  ICLR 2023

2) Semi-supervised Community Detection via Structural Similarity Metrics. By Yicong Jiang and Tracy Ke,  ICLR 2023

3) Random Sparse Lifts: Construction, Analysis and Convergence of finite sparse networks. By David Robin et al., ICLR 2024

Thus, we believe that the network setting presented in this paper is of interest to the machine learning community.

***
**For the second main point**, we have added two numerical experiments  (Experiment 2 and 3 in Section 4) where we compare
R-SCORE with the approach by Ma, Ma, and Yuan (2020). The latter is essentially a non-convex penalized MLE (npMLE) approach for fitting latent space model, which is the work that is most closely related to our work. We compare the error rates of R-SCORE with npMLE under different settings, and find that (a) R-SCORE is computationally much faster, and (b) in many settings, the error rate of
R-SCORE is lower than that of npMLE.

---

### Meta-Review · Area_Chair_grCu · 2024-12-19

**Metareview:**

This paper introduces the logit-DCBM model for random graphs which extends several previous models. The paper then designs R-SCORE to estimate the parameters of the model, which addresses nonlinearities in the model by a cancellation trick, detailed in Lemma 2.2. Empirically R-SCORE seems to outperform existing spectral clustering approaches. Overall the reviewers find the results to be interesting, however there are some concerns about presentation (which was partly addressed in author response) and the significance of the work especially for the ICLR community.

**Additional Comments On Reviewer Discussion:**

Authors clarified the scope and motivation of the setting and addressed some confusions.

---

### Decision · Program_Chairs · 2025-01-22

Accept (Poster)